# A Descriptive Analysis of Transitions from Smoking to Electronic Nicotine Delivery System (ENDS) Use: A Daily Diary Investigation

**DOI:** 10.3390/ijerph18126301

**Published:** 2021-06-10

**Authors:** Tamlin S. Conner, Jiaxu Zeng, Mei-Ling Blank, Vicky He, Janet Hoek

**Affiliations:** 1Department of Psychology, University of Otago, Dunedin 9054, New Zealand; 2Department of Preventive and Social Medicine, University of Otago, Dunedin 9054, New Zealand; jimmy.zeng@otago.ac.nz (J.Z.); meiling.blank@otago.ac.nz (M.-L.B.); vicky.he@otago.ac.nz (V.H.); 3Department of Public Health, University of Otago, Wellington 6242, New Zealand; janet.hoek@otago.ac.nz

**Keywords:** electronic nicotine delivery systems, e-cigarettes, dual use, daily diary

## Abstract

Objectives: We aimed to examine patterns in smoking and electronic nicotine delivery system (ENDS) use over an extended period of time (up to 20 weeks) in people who smoked and who had never previously made a successful quit attempt using an ENDS. Design and setting: We conducted a longitudinal mixed-methods study in Dunedin, New Zealand, during 2018 and 2019. Participants: Purposively selected participants (*N* = 45; age (≥18 years), gender, ethnicities, cigarettes/day) who wished to quit smoking. Interventions: Participants were provided with a second-generation ENDS device (vape pen or starter “tank” device) at the start of their quit attempt, and asked to complete smartphone-based daily diary surveys assessing smoking and ENDS use. Outcome measures: Sunburst plots and a sequence plot were used to describe weekly and daily patterns of smoking and ENDS use (smoking only, ENDS use only, dual use, abstinent). Results: The most frequently reported movements among participants, classified according to their study week behaviour, occurred between dual use and exclusive ENDS use (and vice versa). A smaller group reported moving from dual use to exclusive smoking (and often back to dual use), and a small number reported moving between abstinence and different ENDS and smoked tobacco usage behaviours. Data visualisations focussing on those participants who had provided data during each of weeks 9–12 indicate that only a minority reported sustained dual use; instead, most participants indicated varied smoked tobacco and ENDS use, which included periods of dual use. Conclusions: The considerable variety observed within and between study participants suggests that high variability is typical rather than exceptional. Transitions from smoking to ENDS use may involve considerable periods of dual use, which is likely to be dynamic and potentially sustained over several months.

## 1. Introduction

The rapid adoption of ENDS (electronic nicotine delivery systems) has raised hopes these could prove a “disruptive technology” that accelerates declines in smoking prevalence and potentially ends the smoking epidemic [1,2]. Yet reviews of research into ENDS use and smoking cessation have reached mixed conclusions about ENDS’ potential contribution to public health. For example, a large review and meta-analysis concluded that ENDS use reduced the odds of successful smoking cessation [3], although a Cochrane review of randomised controlled trials found low quality evidence that ENDS containing nicotine could support cessation over six months, relative to ENDS not containing nicotine [4]. An updated Cochrane review of 50 studies, including 26 randomized controlled trials (RCTs), found moderate-certainty evidence that quit rates were higher in people randomized to nicotine ENDS than to nicotine replacement therapy (NRT) [5]. 

Longitudinal studies have also reached varied conclusions; several have reported that ENDS use led to reduced cigarette consumption, [6,7,8,9,10,11] increased sustained quit attempts [6,8,12,13], supported quitting [7,14,15], and militated against relapse to smoking [7,14]. However, others have concluded that ENDS use was not associated with either increased cessation success or reduced tobacco consumption [16,17,18,19], and some found more intensive ENDS use was associated with lower likelihood of cessation [19], and increased risk of relapse [8]. 

Dual use, defined as concurrent use of smoked tobacco and ENDS [20], may help explain these varied outcomes. For example, dual use may enable smokers to become accustomed to a new nicotine-delivery experience and thus support transition to exclusive ENDS use [20]. Alternatively, dual use may become entrenched if smokers use ENDS to manage smokefree restrictions or if they satisfice, in the belief they will benefit by reducing their tobacco consumption even if they do not quit [21,22]. Despite the ambiguous status of dual use and its potential to support or undermine smoking cessation, knowledge of how dual use evolves remains scant.

Of recent longitudinal studies examining dual use patterns one found that, of baseline dual users, 43.9% had moved to exclusive smoking at one year follow-up and 48.8% remained as dual users (the remainder were either exclusive ENDS users or abstinent from both ENDS and smoked tobacco) [23]. An analysis of International Tobacco Control Four Country surveys of smoking and ENDS use found that of baseline daily dual users only, 21.4% were exclusive daily smokers at Wave 2 (18 months later) while 38.5% remained daily dual users and 26.9% moved to non-daily dual use [24]. Among dual users who were predominant smokers (daily or non-daily smoking and less than daily ENDS use), 44.9% had moved to exclusive smoking at Wave 2 while 5.9% of dual users who were predominant ENDS users (daily ENDS use and daily or non-daily smoking) reported exclusive smoking at Wave 2 [24]. 

An analysis of Waves 1 and 2 from the Population Assessment of Tobacco and Health (PATH) study reported that, of Wave 1 dual users, 59.2% were exclusive smokers at Wave 2 (one year later); 25.5% continued dual use and 3.8% reported another smoked tobacco behavior [15]. Similar analyses but using Waves 1 and 3 from the PATH study found that of baseline dual users, 25.7% remained dual users at Wave 3 (two years later) and 55.2% were exclusive smokers (the remainder were either exclusive ENDS users or had stopped using both smoked tobacco and ENDS) [25]. A second study examining Waves 1 and 3 from the PATH dataset analysed movement from dual use to exclusive ENDS use and smoking cessation [26]. Of baseline (Wave 1) dual users, 6.8% had moved to exclusive ENDS use at Wave 3 while 9.3% reported ceasing ENDS and smoked tobacco use [26]. Dual users who were predominant smokers were less likely than other dual use groups to have moved to exclusive ENDS use. 

These studies provide important insights into behaviour changes that occur several months after baseline assessment [9,11,12,13,15]. However, the high variability reported highlights the need for finer-grained analyses of movements from smoking to ENDS use, and the possible continuity and discontinuity in those transitions over time.

Studies using real-time survey approaches such as electronic daily diaries enable more frequent data collection and could potentially characterise how transition from smoking to ENDS use occurs. Smartphones allow frequently administered surveys that probe real-time or recent behaviour, thus potentially reducing recall bias [27]. These approaches show high validity in measuring smoking in daily life and have been shown to correspond with objective biomarkers of smoking, such as salivary cotinine levels [28,29]. Moreover, smartphones represent an effective survey medium as they have high penetration across diverse population groups. Phone owners typically use and check their devices several times a day, and many give incoming messages and calls priority over other activities. The longer survey format afforded by a once-daily diary approach could enable collection of more varied ENDS use behaviours, including the frequency and length of ENDS use sessions, and the number of puffs taken [30,31,32].

Recent work using ecological momentary assessment (EMA) and daily diaries has examined subjective experiences of ENDS and cigarette use [30,33], examined how variations in nicotine levels affect usage [29,30,34], and documented co-use of psycho-active products [33]. However, these studies typically involved intensive data collection over a relatively short period of time (two to three weeks) [30], or more intensive intervention [29]. We are not aware of any studies using EMA or daily diary approaches to characterise how transitions from smoking to ENDS occur naturally over an extended time period.

This knowledge gap is important as more detailed descriptions of ENDS uptake and use, and cigarette consumption, could guide practice. We therefore examined patterns in smoking and ENDS use over an extended period of time (up to 20 weeks) in people who smoked and who had never previously made a successful quit attempt using an ENDS. We provided participants with an ENDS and then used smartphone-based daily diary surveys to examine their reported ENDS and smoked tobacco use. Our study responds to calls for more individual-based analyses [26], and the detailed transition patterns we describe complement existing longer-term studies that use extended follow-up periods.

## 2. Methods

### 2.1. Smoking-to-Vaping Study Overview

The Smoking-to-Vaping (S2V) Study was a longitudinal mixed-methods study conducted in Dunedin, New Zealand (NZ) from May–December 2018 and March–September 2019. Full details of the study, including ENDS regulation in NZ during the study period, are available in the methods report [35]. Eligible participants were at least 18 years old, had smoked at least 100 cigarettes in their lifetime, were smoking at least one cigarette per week, were not currently using an ENDS once a week or more often, were not currently trying to quit using any means, had never used an ENDS to stop smoking for 30 days or more, were willing to use an ENDS to try and quit smoking, had access to an Internet-connected smartphone, and were willing to answer short online surveys every day. Exclusion criteria included current pregnancy or breastfeeding, recently diagnosed major health condition (e.g., heart attack, cancer, serious psychiatric illness), poorly controlled respiratory disease (e.g., severe asthma), and a household member already participating in the study. We purposively sampled eligible participants to maximize sample diversity (age, gender, ethnicities, number of cigarettes smoked per day). 

Participants attended up to five in-depth interviews (not analysed in this manuscript) over 18–20 weeks and received a text message every day at 10 AM (and a reminder) requesting them to complete a short survey (1–2 min) about their smoking and ENDS use for the previous day, to capture the entire day’s activities. A 24-h recall period is generally regarded as acceptable for assessing health behaviours [36]. To encourage survey response, participants received additional reimbursement if they completed at least 50% of the surveys since their last interview (maximum reimbursement NZ$260 (2018) or NZ$290 (2019) for interview attendance and daily survey completion). 

All participants chose an ENDS starter kit (value up to NZ$80) from a specialist ENDS retail store during their intake session and purchased their own e-liquids during the study. Participants reported using varied e-liquid flavours throughout the study period [37]; the nicotine concentrations in e-liquids purchased varied within and between participants (from 0 mg to ~50 mg). Participants did not receive behavioural cessation support from the research team and were considered lost to follow-up if they did not respond to at least two researcher-initiated contacts to reschedule missed interviews.

### 2.2. Measures

#### 2.2.1. Daily Survey—Smoking and ENDS Use

Participants were asked if yesterday they had at least one puff on a cigarette (yes/no), and if yes, how many cigarettes they smoked (at least one puff) (1–“more than 60”). Participants were then asked if yesterday they used their ENDS (even just one puff) (yes/no), and if yes, how many times they had at least one puff (1 time, 2–5 times, 6–10 times, 11–15 times, 16–20 times, 21–25 times, 26–30 times, 31 or more times).

With these data, we created a record of smoking, and ENDS use, by “study day”, and “study week”, for each participant. Specifically, we classified participants’ study days, and study weeks as: “smoking only”, “ENDS use only”, “dual use” (smoking and ENDS use) or “abstinent” (no smoking or ENDS use), as described below.

#### 2.2.2. Smoking and ENDS Use, by “Study Day” 

We operationalised the definitions outlined above to classify each participant’s smoking and ENDS use for every day they were enrolled in the study: if a participant reported at least one puff of a cigarette and no ENDS use, that day was classified as “smoking only”. If they reported at least one ENDS puff and no smoking, that day was classified as “ENDS use only”. If they reported at least one puff of a cigarette and at least one ENDS puff, that day was classified as “daily dual use”. Finally, if they reported no smoking and no ENDS use, that day was classified as “abstinent”. Study days with no responses were classified as “missing”.

#### 2.2.3. Smoking, and ENDS Use, by “Study Week”

We also categorised participants’ enrolled time in the study by “study week” (e.g., the first seven days of a participant’s enrolment was “week 1”, etc.). We then used the available “study day” smoking and ENDS use data to classify each participant’s “study weeks”. For example, if, during a specific study week, participants reported at least one study day smoking and no days of ENDS use, that week was classified as “smoking only”. If they reported at least one day of ENDS use and no days of smoking, that week was classified as “ENDS use only”. If they reported at least one day smoking and one day of ENDS use (occurring on the same study day or on separate days of the study week), that week was classified as “weekly dual use”. Finally, if they did not report any days of smoking or ENDS use, that week was classified as “abstinent”. Study weeks with no study day responses were classified as “missing”.

### 2.3. Classification of Smoking and ENDS Use Transitions

We classified transitions for only those participants who both (a) were enrolled and provided at least some survey responses for at least 12 weeks from their intake session, and (b) had “complete” study week data for each of the weeks 9–12. We used a 12 week enrolment threshold as smoking cessation programs commonly provide support for 12 weeks [38]. Most importantly, classifying transitions for each participant only after a similar elapsed time within the study avoided introducing bias due to differing lengths of follow-up time. We classified transitions based on weeks 9–12 (i.e., assessing “study week”-level data for weeks 9, 10, 11 and 12 only) as many clinical guidelines define “ex-smokers” as not having had a single puff in the past month [39]. Categorising the study cohort (*N* = 45) based on these two criteria, 31 participants were included in the transition classification analysis. Of the participants excluded from the analysis (*n* = 14), most withdrew or were lost to follow-up within 12 weeks of their intake session, or remained in the study by attending follow-up interviews but stopped responding to survey requests (*n* = 13); one had “incomplete” study week level data for each of the weeks 9–12.

For the transition classification analysis, participants were classified as “smoking only” if all four weeks of the week 9–12 period indicated they only smoked and never used an ENDS. They were classified as “ENDS use only” if all four weeks indicated they only used an ENDS, and never smoked, during that period. They were classified as “dual use” if any of their week 9–12 data indicated “weekly dual use”, or if there was any combination of alternating “smoking only” and “ENDS use only” study weeks during that period. They were classified as “abstinent” if all four weeks of the week 9–12 period indicated no smoking and no ENDS use.

### 2.4. Smoking, ENDS Use and Demographic Covariates

Covariates included smoking history (age at first cigarette, age at first weekly cigarette smoking, tobacco type [tailor-made cigarettes, roll-your-own tobacco, both], ever-tried to quit), ENDS use history (ever-tried ENDS, ever-owned ENDS), confidence in quitting (continuous, 0–100), and demographics (ethnicity [Māori, New Zealand European, Other Asian], education [high school or below, above high school]). 

### 2.5. Statistical Analyses 

Descriptive analyses compared the baseline characteristics of the participants included and excluded from the transition classification analysis. For those included in the analysis, descriptive analyses compared the baseline characteristics of these participants according to their transition classification after 12 weeks of enrolment. We constructed Sunburst Plots to summarise and compare the “study week”-level smoking and ENDS use patterns for all enrolled participants, those included in the transition analysis, and those excluded from the analysis. We used Sequence Plots constructed from the “study day”-level data to describe all of the enrolled participants’ smoking and ENDS usage for their entire study enrolment. Descriptive analyses were conducted in Stata/IC v16.1 (StataCorp, College Station, TX, USA); the sunburst and sequence plots were constructed in R v.3.6.3 (R Core Team, Vienna, Austria, 2020).

## 3. Results

### 3.1. Participant Characteristics

Forty-five participants enrolled in the study, participated for a median of 18 weeks (IQR = 8), and completed a median of 75 daily surveys each (IQR = 84, 60% overall response rate). Table 1 shows the baseline characteristics for all enrolled participants (*N* = 45), those included in the transition classification analysis (*n* = 31), and those excluded (*n* = 14). Among all enrolled participants, most were aged between 23–47 years (*Mdn* = 32), female (*n* = 26, 57.8%), NZ European (*n* = 30, 66.7%), or Māori (indigenous peoples of NZ; *n* = 13, 28.9%) ethnicities, and had completed high school or below (*n* = 25, 55.6%). Most had tried to quit smoking previously (*n* = 34, 75.6%), and had previously tried ENDS (*n* = 28, 62.2%), though few had ever owned an ENDS (*n* = 6, 13.3%). The baseline characteristics of those included in the transition analysis differed substantially from those excluded. Included participants were more likely to be male, NZ European, have greater than a high school education, and smoke roll-your-own tobacco. Participants left the study at different time points; half (*n* = 23) were still contributing daily survey responses at week 18 (*n* = 6 at week 20).

### 3.2. Transition Classifications and Baseline Characteristics

Table 2 describes the transition classification of the 31 participants included in the analysis according to their baseline characteristics (note, the table presents row percentages). Of the 31 participants included in the transition classification analysis (sunburst plot panel B, based on study weeks 9–12, indicated by the dashed line), most were classified as “dual users” (*n* = 21, 67.7%); nine (29.0%) were classified as “ENDS use only”; one (3.3%) was classified as “smoking only”. None were classified as “abstinent”. A higher proportion of those who were of NZ European ethnicity, first smoked at age 13–19, ever-tried to quit smoking, and never owned an ENDS, were classified as “ENDS use only”.

### 3.3. Sunburst Plots

The sunburst plots show the usage patterns by “study week” for all enrolled participants (Figure 1, Panel A), those included in the transition classification analysis (Figure 1, Panel B), and those excluded (Figure 1, Panel C).

The sunburst plots presented here are similar to pie charts, with the added dimension of time. Each “ring” represents a “study week” (the innermost ring represents study week 1) and summarises the proportion of participants according to the classification categories (smoking only, ENDS use only, weekly dual use, abstinent, missing data) for that specific week. The plots allow us to visualise the complexity of the transition process and the difficulty in assigning transition “outcomes” during an inherently unstable, non-linear process.

For example, looking at the innermost ring (study week 1) of Panel A, we can see that almost all participants were classified as “weekly dual use” (orange), with a small proportion classified as “ENDS use only” (yellow). Moving to the next ring (study week 2), a substantial proportion of the “weekly dual use” participants maintained this pattern. However, some of these participants were classified as “ENDS use only” at week 2, a smaller proportion was classified as “smoking only” (blue), and the smallest proportion was classified as “missing data” (light blue). Among the participants classified as “ENDS use only” at week 1, half were classified as “ENDS use only” and half as “weekly dual use” at week 2. Hence, the plots show the dynamic switching between usage categories that occurs over time.

Among participants included in the transition analysis (*n* = 31, Panel B), nearly all rapidly adopted a “weekly dual use” pattern; a large proportion maintained this pattern throughout their participation. However, we also observed some of these participants transition to “ENDS use only”, relapse to “smoking only”, or switch between different smoking and ENDS use patterns until study exit. After the 12 week period used for the transition classification, there was considerable switching between “weekly dual use” and “ENDS use only”, and relatively few instances where participants abstained from both smoking and ENDS use.

Among those excluded from the analysis (*n* = 14, Panel C), all showed “weekly dual use” in the first week; about half had at least one week classified as “smoking only” within the first six weeks, indicating that these participants relapsed to smoking more quickly than those included in the transition analyses. By study week eight, only three participants responded to any survey requests.

### 3.4. Sequence Plot

Figure 2 is a sequence plot constructed from “study day”-level data for each enrolled participant (*N* = 45). The plot contains 45 rows; each row reflects patterns for one participant. Similar to the “study week”-level sunburst plot, most participants began as “daily dual users” but then developed divergent usage patterns. Generally, the proportion of participants classified as “daily dual use” (shown in orange) tapered off, and the proportion classified as “ENDS use only” (yellow) increased, although this pattern was often interspersed with study days of “daily dual use”. Some participants resumed “smoking only” (blue) either relatively quickly, or after longer periods of “daily dual use”; relatively few abstained (green) from both smoking and ENDS use. Participant non-responses mean that Figure 2 has considerable missing data.

## 4. Discussion

People attempting to switch from smoking to ENDS use reported highly varied smoking and ENDS use behaviours. The most frequently reported movements among participants classified according to their study week behaviour occurred between dual use and exclusive ENDS use (and vice versa). A smaller group reported moving from dual use to exclusive smoking (and often back to dual use), and a small number reported moving between abstinence and different ENDS and smoking behaviours. Data visualizations focussing on those participants who had provided data during each of weeks 9–12 indicate that only a minority reported sustained dual use; instead, most participants indicated varied smoking and ENDS use, which included periods of dual use. The considerable variety observed within and between study participants suggests high variability is typical rather than exceptional.

Visualization of usage patterns at the daily level showed considerable alternation between daily dual use and ENDS use only. This pattern may indicate participants tested whether they could successfully substitute ENDS use for smoking but had difficulty developing ENDS use practices that effectively replaced smoking. Qualitative analyses of dual use suggest stress and peer group norms may affect ENDS use within and between days [22,40,41]; people attempting to switch from smoking to ENDS use may thus benefit from advice that variability is normal. 

Dual use has often been viewed as a transitional phase through which all smokers pass before adopting a harm reduction approach (i.e., switching to exclusive ENDS use) or stopping use of all nicotine products (abstinence) [26]. Only more recently have researchers questioned whether dual use could reduce successful smoking cessation, by becoming established as a sustained behaviour, increasing nicotine dependence, or pre-disposing relapse to exclusive smoking [3]. Our findings reinforce earlier studies that found dual use could continue, or be followed by exclusive smoking, exclusive ENDS use, or abstinence [23,24,25,26]. Yet unlike the ITC study that followed participants for 18 months and reported 12% of exclusive smokers (daily and non-daily) at baseline were dual using at follow-up [24], we found that, at week 12, 68% of participants (*n* = 21) included in the transition classification analysis were classified as weekly dual users. 

This difference may reflect the longer follow-up periods used in these studies and the difficulty of remaining smokefree, even after a sustained period during which participants had not smoked [42]. It may also indicate increased nicotine dependence; recent research found that although dual users reported smoking fewer cigarettes, their nicotine use and dependence increased [43]. Because smoking typically provides more rapid nicotine delivery than ENDS use [44], dual users with increased nicotine dependence may revert to exclusive smoking. Our small sample does not allow investigation of this latter possibility but future longitudinal studies could explore how nicotine dependence influences the evolution of dual use.

The fine-grained analyses we undertook extend current understanding of smoking and ENDS use and have important implications. First, as shown in the Sunburst Plots, after week 12 a large proportion of participants exhibited unstable behaviours, switching between dual use, smoking only, and ENDS use only. Furthermore, some participants classified as dual users returned to smoking only, and one participant classified as ENDS use only became abstinent. These findings suggest classifying dynamic behaviours such as smoking and ENDS use using point prevalence categories (e.g., “exclusive smoking”) may be overly simplistic, especially during the early stages of an ENDS-assisted quit attempt. Treating cigarette and ENDS consumption over time as continuous variables would avoid use of binary classification measures and enable modelling of changes in smoking and ENDS use. Future research employing passive data collection methods (e.g., sensor-enabled wearables) could implement this suggestion; this approach would also avoid measurement error (e.g., recall bias) that might arise from self-reported data. Second, the considerable volatility we observed as participants moved between dual use and exclusive ENDS use suggests smokers hoping to transition to exclusive ENDS use could benefit from advice that encourages persistence, particularly if progress is slower than desired, to ameliorate the risk of relapse to exclusive smoking. Third, evidence that people move in and out of dual use suggests cessation support programmes involving ENDS may need to run for longer than a conventional 12-week period to ensure people requiring extended behavioural support receive it. Fourth, the approach we have used could examine movement from exclusive ENDS use to ceasing use of any nicotine product. While ENDS are generally regarded as less harmful than smoking, they are not risk-free [45,46,47]; describing pathways from exclusive ENDS use to abstinence could increase understanding of how this transition occurs. 

At a population level, dual use presents health risks. Preliminary cross-sectional research has linked regular dual use of combustible cigarettes and e-cigarettes to cardiopulmonary health problems including greater self-reported breathing difficulties [48], and greater self-reported cardiovascular disease compared to cigarette-only users [49], although other research shows equivalent vascular impairment among cigarette smokers, sole ENDS users, and dual users [46]. Further research is needed to clarify harm from dual use versus sole-use of ENDS. 

Our study has limitations, including the small sample size and variable response rates between participants to the daily surveys; therefore we described patterns of smoking and ENDS use only. To recognise the impact of participant withdrawal/loss to follow-up, we present separate analyses of participants who did, and did not, report data for each of weeks 9–12; ideally, however, all participants would have provided 18–20 weeks of data. Furthermore, comparing the baseline characteristics of those included and excluded from the transition classification analysis revealed systematic differences; excluded participants were more likely to be female, have lower educational attainment, and return to exclusive smoking more quickly. Smoking and ENDS use were self-reported with no biochemical verification, and we lacked a true baseline measure of cigarette consumption prior to receiving the ENDS. Because our interest lay in smoked tobacco and ENDS use, we did not explore other practices, such as marijuana use; given calls to legalise marijuana use in New Zealand, future work could examine both tobacco and marijuana consumption. Nor did we collect details of participants’ e-liquid flavour use or nicotine concentrations via the daily diary survey (though we explored both topics in the qualitative interviews). Lastly, despite repeated advice that the study was not a cessation study, some participants interpreted the regular in-depth interviews as behavioural support.

Nonetheless, our study has important strengths; the intensive 18–20 week daily diary surveys enabled an in-depth exploration of smoking and ENDS use over time and potentially minimised recall bias that may affect retrospective studies. The data visualizations are, to our knowledge, the first to support such granular depictions of ENDS and smoking over time. Providing participants with a starter ENDS device removed financial barriers to participation and sample members varied by age, gender and education, and included 13 Māori participants. 

In summary, transitions from smoking to ENDS use are likely to involve considerable dual use, which is likely to be dynamic and potentially sustained over several months. Advising people who wish to switch to exclusive ENDS use to view this variability as typical, and to persist in their attempts to stop smoking, even if their progress feels slow and uncertain, may reframe how they approach ENDS uptake. Managing expectations while sustaining self-efficacy and motivation to quit may benefit from sustained cessation support. Further research examining factors associated with successful transition could fine-tune this advice, potentially enable more straightforward transitions, and assist people to move from ENDS-use to abstinence.

### Strengths and Limitations of This Study


This study is the first to use an intensive longitudinal approach to examine patterns of smoking and ENDS use among a group of adult smokers.The daily diary survey method enabled a more fine-grained description of smoking and ENDS use than those reported in previous studies.The small sample of participants was purposively selected and may not be representative of New Zealand smokers.Smoking and ENDS use were self-reported, and we lacked a true baseline measure of cigarette consumption before study enrolment.High levels of participant withdrawal and loss to follow-up limited the sample used in the transition classification analysis.


## Figures and Tables

**Figure 1 ijerph-18-06301-f001:**
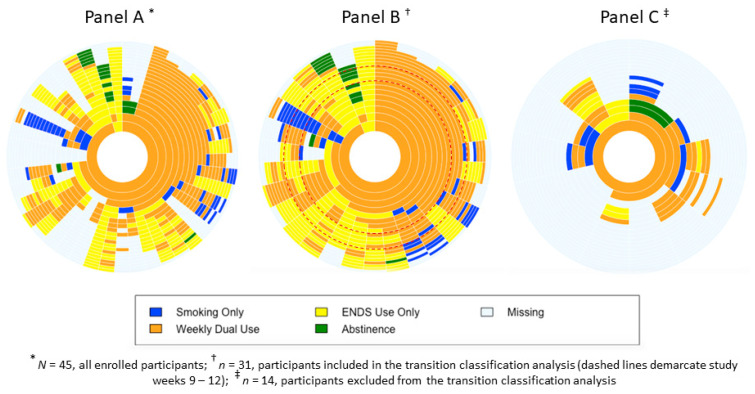
Sunburst plots of smoking and ENDS usage by weeks.

**Figure 2 ijerph-18-06301-f002:**
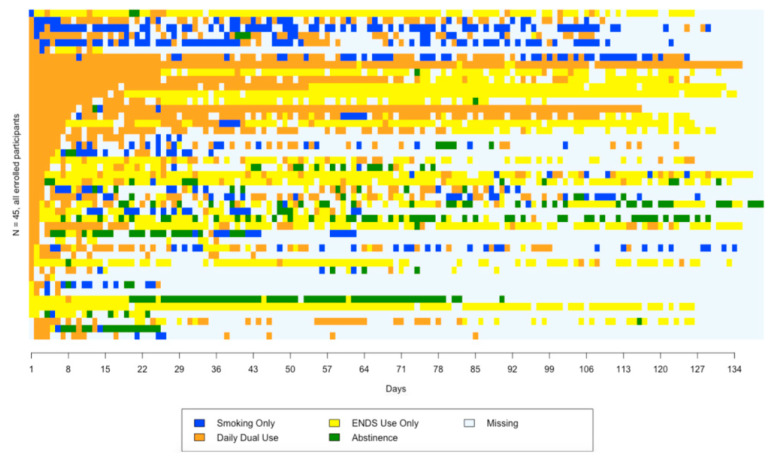
Sequence plot of smoking and ENDS usage by days.

**Table 1 ijerph-18-06301-t001:** Baseline characteristics of all enrolled Smoking-to-Vaping study participants (*N* = 45), those included in the transition classification analysis (*n* = 31) *, and those excluded from the transition analysis (*n* = 14).

	All Enrolled Participants *N* = 45	Included in the Transition Classification Analysis * *n* = 31	Excluded from the Transition Classification Analysis *n* = 14
**Age**, median (Q_1_ to Q_3_) ^†^	32 (23 to 47)	37 (26 to 50)	25 (20 to 38)
**Sex**, *n* (%)			
Male	19 (42.2)	15 (48.4)	4 (28.6)
Female	26 (57.8)	16 (51.6)	10 (71.4)
**Prioritised ethnicity**, *n* (%)			
Māori	13 (28.9)	8 (25.8)	5 (35.7)
New Zealand European	30 (66.7)	22 (71.0)	8 (57.2)
Other Asian	2 (4.4)	1 (3.2)	1 (7.1)
**Education**, *n* (%)			
Above high school	20 (44.4)	15 (48.4)	5 (35.7)
High school or below	25 (55.6)	16 (51.6)	9 (64.3)
**Age at first puff of cigarette (years)**, *n* (%)			
<13	15 (33.3)	10 (32.3)	5 (35.7)
≥13 and <19	30 (66.7)	21 (67.7)	9 (64.3)
≥19	0 (0)	0 (0)	0 (0)
**Age at first weekly cigarette smoking (years)**, *n* (%)			
<13	4 (8.9)	2 (6.5)	2 (14.3)
≥13 and <19	37 (82.2)	25 (80.6)	12 (85.7)
≥19	4 (8.9)	4 (12.9)	0 (0)
**Tobacco type**, *n* (%)			
Roll-your-own	15 (33.3)	14 (45.2)	1 (7.1)
Tailor-made	15 (33.3)	11 (35.5)	4 (28.6)
Both	15 (33.4)	6 (19.4)	9 (64.3)
**Ever-tried to quit smoking**, *n* (%)			
Yes	34 (75.6)	24 (77.4)	10 (71.4)
No	11 (24.4)	7 (22.6)	4 (28.6)
**Ever-tried ENDS**, *n* (%)			
Yes	28 (62.2)	19 (61.3)	9 (64.3)
No	17 (37.8)	12 (38.7)	5 (35.7)
**Ever-owned ENDS**, *n* (%)			
Yes	6 (13.3)	4 (12.9)	2 (14.3)
No	39 (86.7)	27 (87.1)	12 (85.7)
**Confidence in quitting smoking**, median (Q_1_ to Q_3_)	78 (60 to 84)	79 (60 to 90)	70.5 (60 to 80)

* Only those participants who both (a) were enrolled and provided at least some survey responses for at least 12 weeks from their intake session, and (b) had “complete” study week data for each of the weeks 9–12. ^†^ Q_1_ and Q_3_ are the lower and upper quartiles, respectively.

**Table 2 ijerph-18-06301-t002:** Baseline characteristics of the participants included in the transition classification analysis (*n* = 31) *, based on their transition classification after 12 weeks of study enrolment.

	Dual Use *n* = 21 (67.7%)	ENDS Use Only *n* = 9 (29.0%)	Smoking Only *n* = 1 (3.3%)
**Age**, median (Q_1_ to Q_3_) ^†^	36 (23 to 49)	40 (31 to 49)	37 (NA)
**Sex**, *n* (row %)			
Male	11 (73.3)	4 (26.7)	0 (0)
Female	10 (62.5)	5 (31.3)	1 (6.2)
**Prioritised ethnicity**, *n* (row %)			
Māori	6 (75.0)	2 (25.0)	0 (0)
New Zealand European	14 (63.6)	7 (32.8)	1 (4.6)
Other Asian	1 (100)	0 (0)	0 (0)
**Education**, *n* (row %)			
Above high school	9 (60.0)	5 (33.3)	1 (6.7)
High school or below	12 (75.0)	4 (25.0)	0 (0)
**Age at first puff of cigarette (years)**, *n* (row %)			
<13	7 (70.0)	3 (30.0)	0 (0)
≥13 and <19	14 (66.7)	6 (28.6)	1 (4.7)
≥19	0 (0)	0 (0)	0 (0)
**Age at first weekly cigarette smoking (years)**, *n* (row %)			
<13	1 (50.0)	1 (50.0)	0 (0)
≥13 and <19	18 (72.0)	6 (24.0)	1 (4.0)
≥19	2 (50.0)	2 (50.0)	0 (0)
**Tobacco type**, *n* (row %)			
Roll-your-own	10 (71.4)	4 (28.6)	0 (0)
Tailor-made	7 (63.6)	3 (27.3)	1 (9.1)
Both	4 (66.7)	2 (33.3)	0 (0)
**Ever-tried to quit smoking**, *n* (row %)			
Yes	17 (70.8)	6 (25.0)	1 (4.2)
No	4 (57.1)	3 (42.9)	0 (0)
**Ever-tried ENDS**, *n* (row %)			
Yes	13 (68.4)	5 (26.3)	1 (5.3)
No	8 (66.7)	4 (33.3)	0 (0)
**Ever-owned ENDS**, *n* (row %)			
Yes	2 (50.0)	1 (25.0)	1 (25.0)
No	19 (70.4)	8 (29.6)	0 (0)
**Confidence in quitting smoking**, median (Q_1_ to Q_3_)	78 (68 to 88)	84 (73 to 94)	50 (NA)

* Only those participants who both (a) were enrolled and provided at least some survey responses for at least 12 weeks from their intake session, and (b) had “complete” study week data for each of the weeks 9–12. ^†^ Q_1_ and Q_3_ are the lower and upper quartiles, respectively.

## Data Availability

The datasets generated and analysed during the current study are available from the corresponding author on reasonable request.

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
