# Peer review of "A Descriptive Analysis of Transitions from Smoking to Electronic Nicotine Delivery System (ENDS) Use: A Daily Diary Investigation"

_ijerph, 2021, doi:10.3390/ijerph18126301_

Round 1

Reviewer 1 Report

Thank you for asking me to review this manuscript. In the present longitudinal study, Conner and colleagues examined the patterns in smoking and electronic nicotine delivery system (ENDS) use over an extended period of time (up to 20 weeks) in people who smoked and who had never previously made a successful quit attempt using an ENDS.

The authors underline the considerable variety observed within and between study participants suggesting that high variability is typical rather than exceptional.

Indeed, transitions from smoking to ENDS use may involve considerable periods of dual use, which is likely to be dynamic and potentially sustained over several months. The paper is very fluent, clear, and easy to read. Overall, I think that the manuscript fits within the scope of this journal on an important topic and the data are very interesting even if future larger, randomized, controlled studies are warranted.

I make some suggestions for improve the overall quality of paper.

  • The authors should better justify the sample size and eventually need to do a power calculation. It is a crucial point to address in this type of study design.
  • Please better focus on possible mechanism of action (for example… role of nicotine receptors) underlying the results here presented. I guess that the discussion should be further improved and extended in this context.
  • Based on these results, what is the connection between smoking/ENDS use and weight loss or weight gain? Please make an appropriate comment in discussion section of manuscript and insert appropriate references.

Author Response

Reviewer 1

In the present longitudinal study, Conner and colleagues examined the patterns in smoking and electronic nicotine delivery system (ENDS) use over an extended period of time (up to 20 weeks) in people who smoked and who had never previously made a successful quit attempt using an ENDS.

The authors underline the considerable variety observed within and between study participants suggesting that high variability is typical rather than exceptional.

Indeed, transitions from smoking to ENDS use may involve considerable periods of dual use, which is likely to be dynamic and potentially sustained over several months. The paper is very fluent, clear, and easy to read. Overall, I think that the manuscript fits within the scope of this journal on an important topic and the data are very interesting even if future larger, randomized, controlled studies are warranted.

I make some suggestions for improve the overall quality of paper.

  1. The authors should better justify the sample size and eventually need to do a power calculation. It is a crucial point to address in this type of study design.

Please see our response to the SE in point 5, above.

  1. Please better focus on possible mechanism of action (for example… role of nicotine receptors) underlying the results here presented. I guess that the discussion should be further improved and extended in this context.

Although we appreciate this suggestion, including comments to this effect would go well beyond the data we have collected and the research questions we address. None of the authors is qualified to discuss biochemical pathways and we feel this discussion would be better situated in a study that has explicit measures of nicotine uptake (i.e., via blood samples taken in laboratory sessions).

  1. Based on these results, what is the connection between smoking/ENDS use and weight loss or weight gain? Please make an appropriate comment in discussion section of manuscript and insert appropriate references.

Again, while we understand the importance of examining psycho-social factors that may predispose and sustain nicotine use (either via smoking or ENDS), we feel this suggestion goes well beyond the data we have collected and the research questions we set out to address.

Reviewer 2 Report

I believe it is an interesting attempt to study the possibilities of smokers trying to quit.

(1). However, care must be taken to warn the population that the double use - tobacco cigarettes and e-cigarettes - can be dangerous and complicate life even more, as it would be exposed to two different types of nicotine release, or even e-cigarette don't have nicotine.

(2). I also believe that, this as a private opinion, because it is dealing with a certain number of people, that the study could be accompanied by brochures on how nicotine creates addiction and how it affects the lungs.

(3). Showing, for example, X-ray radiography of healthy lungs and smokers, so that they check the degree of deterioration that happens due to the exposure of smoke in the pulmonary area.

(4). On the other hand, to make it known how, mainly men can lose their sexual potency, and in all individuals, the decrease of the disposition to the activities in general.

Author Response

Reviewer 2

I believe it is an interesting attempt to study the possibilities of smokers trying to quit.

  1. However, care must be taken to warn the population that the double use - tobacco cigarettes and e-cigarettes - can be dangerous and complicate life even more, as it would be exposed to two different types of nicotine release, or even e-cigarette don't have nicotine.

We agree that dual use carries risks, given dual users by definition continue using combusted tobacco. We think it is important to differentiate between nicotine and the products of combustion, given it is the latter that cause harm and, by comparison, nicotine poses far fewer risks. Please see pages 8-9, where we comment on risks of dual use “Dual use has often been viewed as a transitional phase through which all smokers pass before adopting a harm reduction approach (i.e., switching to exclusive ENDS use) or stopping use of all nicotine products (abstinence). Only more recently have researchers questioned whether dual use could reduce successful smoking cessation, either by becoming established as a sustained behaviour, increasing nicotine dependence, or by pre-disposing relapse to exclusive smoking. Our findings reinforce earlier studies that found dual use could continue, or be followed by exclusive smoking, exclusive ENDS use, or abstinence.”

We have also added a brief paragraph in the discussion about health risks of dual use (see Special Issue Editor’s Comments point 6):   “At a population level, dual use presents health risks. Preliminary cross-sectional research has linked regular dual use of combustible cigarettes and e-cigarettes to cardiopulmonary health problems including greater self-reported breathing difficulties,49 and greater self-reported cardiovascular disease compared to cigarette-only users,50 although other research shows equivalent vascular impairment among cigarette smokers, sole ENDS users, and dual users.47  Further research is needed to clarify harm from dual use versus sole-use of ENDS.” 

  1. I also believe that, this as a private opinion, because it is dealing with a certain number of people, that the study could be accompanied by brochures on how nicotine creates addiction and how it affects the lungs.

As we noted in our methods section, the study did not offer participants cessation support. At the study end, we provided participants with information they could use to self-refer to cessation support, if they wished.

  1. Showing, for example, X-ray radiography of healthy lungs and smokers, so that they check the degree of deterioration that happens due to the exposure of smoke in the pulmonary area.

While we understand that exposure to this information could cue smoking cessation and help maintain smokefree behaviour, we feel these suggestions go beyond our study, which was not designed as a cessation trial.

  1. On the other hand, to make it known how, mainly men can lose their sexual potency, and in all individuals, the decrease of the disposition to the activities in general.

We agree erectile dysfunction is an important risk factor caused by smoking; however, given our research questions focussed on describing smoking and ENDS use, we feel this suggestion goes beyond what we can legitimately suggest based on our data.